# Using RNA Interference to Reveal the Function of Chromatin Remodeling Factor ISWI in Temperature Tolerance in *Bemisia tabaci* Middle East–Asia Minor 1 Cryptic Species

**DOI:** 10.3390/insects11020113

**Published:** 2020-02-10

**Authors:** Shun-Xia Ji, Xiao-Di Wang, Xiao-Na Shen, Lin Liang, Wan-Xue Liu, Fang-Hao Wan, Zhi-Chuang Lü

**Affiliations:** 1State Key Laboratory for Biology of Plant Diseases and Insect Pests, Institute of Plant Protection, Chinese Academy of Agricultural Sciences, Beijing 100193, China; 82101172334@caas.cn (S.-X.J.); wxd203741084@163.com (X.-D.W.); naxiaoshen@163.com (X.-N.S.); 13808790897@163.com (L.L.); liuwanxue@caas.cn (W.-X.L.); wanfanghao@caas.cn (F.-H.W.); 2Agricultural Genome Institute at Shenzhen, Chinese Academy of Agricultural Sciences, Shenzhen 518120, China

**Keywords:** *Bemisia tabaci*, epigenetics, ISWI, RNA interference, temperature preference, thermal stress

## Abstract

Invasive species often encounter rapid environmental changes during invasions and only the individuals that successfully overcome environmental stresses can colonize and spread. Chromatin remodeling may be essential in environmental adaptation. To assess the functions of imitation switch (ISWI) in invasive *Bemisia tabaci* Middle East–Asia Minor 1 (MEAM1) cryptic species, we cloned and characterized the MEAM1 *BtISWI* gene and determined its functions in response to thermal stress. The full-length cDNA of *BtISWI* was 3712 bp, with a 3068 bp open reading frame (ORF) encoding a 118.86 kDa protein. *BtISWI* mRNA expression was significantly up-regulated after exposure to heat shock or cold shock conditions, indicating that *BtISWI* expression can be induced by thermal stress. After feeding double-stranded RNA (dsRNA), specifically for *BtISWI*, resistance to both heat and cold decreased significantly, suggesting that *BtISWI* may function directly in the thermal tolerance of MEAM1. Moreover, the preferred temperature of MEAM1 adults fed dsRNA was 1.9–3.5 °C higher than the control groups. Taken together, our findings highlight the importance of epigenetic gene regulation in the thermal response or thermal adaptation of invasive *Bemisia tabaci* (*B. tabaci*), and provide a new potential target for establishing sustainable control strategies for *B. tabaci*.

## 1. Introduction

Epigenetics is attracting increasing research attention because it has rapid and long-lasting effects on gene expression in response to environmental changes without altering DNA sequences [1,2]. Epigenetic regulation mediates phenotypic changes caused by various stresses in numerous organisms [3]. As invasive species often encounter rapid environmental changes during invasions, only the individuals successfully overcoming environmental stresses will colonize and spread, and they provide an ideal model to investigate how species epigenetically cope with various stressful environments [4,5,6,7]. Previous studies have suggested that epigenetic regulation may function directly in environmental adaptation, and may be particularly important for the success of invasive species [8,9,10,11].

Adenosine triphosphate (ATP)-dependent chromatin remodeling factors are important components in epigenetic regulation. These factors are multisubunit protein complexes that utilize the energy generated from ATP hydrolysis to modulate the access of transcription factors (TFs) and other regulatory proteins to genomic DNA by sliding nucleosomes and increasing chromatin fluidity [12,13]. According to the conserved domains, these remodelers were divided into four subfamilies: SWI/SNF (SWItch/Sucrose Non Fermentable), ISWI (Imitation SWItch), INO80 (INOsitol requiring 80), and CHD (Chromo Helicase Domain) [14,15,16]. INO80 and CHD are large complexes containing many subunits, while the ISWI complexes are relatively small and contain only two, three, or four subunits [17]. In addition, the number of ISWI-containing complexes is unequal in different organisms. Six different functional ISWI complexes have been identified (CHRAC(chromatin accessibility complex), ACF(ATP-utilizing chromatin assembly and remodeling factor), NURF (nucleosome remodeling factor), RSF(remodeling and spacing factor), ToRC (toutatis containg chromatin remodeling complex), and NoRC(nucleolar remodeling complex)) in *Drosophila melanogaster* [18,19], while seven different ISWI complexes have been described (NURF, ACF, WICH (WSTF-ISWI chromatin remodeling complex), NoRC, CHRAC, RSF, and CERF(CECR2-containing remodeling factor)) in mammals [20].

ISWI-containing protein complexes have multiple roles. For example, research in *Drosophila* has shown that NURF, a chromatin-remodeling complex that contains ISWI as its ATPase subunit, cooperates with the GAGA factor (GAGA-DBD) to mobilize nucleosomes on the promoter of heat-shock genes and facilitate the binding of heat-shock TFs [21,22]. In yeast, ISWI also played an important role in regulating heat shock genes [23]. Buszewicz et al. [24] found that chromatin remodeling complex and histone deacetylase interacted and both were involved in mediating the heat stress response in *Arabidopsis*. Genetic studies in *Saccharomyces cerevisiae* suggested that ISWI complexes are involved in the transcriptional repression of early meiotic genes [25]. Research into plants showed that ISWI ATPase acted as a repressor of at least two flowering pathways [26]. In addition, Collins and co-workers found that ISWI in mammals was required for DNA replication through highly condensed heterochromatin [27].

The whitefly *Bemisia tabaci* (Gennadius) (Hemiptera: Aleyrodidae) is a globally important pest in agriculture that causes damage to many crops through phloem feeding, transmission of plant viruses, and deposition of honeydew [28]. It is considered to be a species complex that contains at least 36 morphologically indistinguishable species [29,30,31]. The *B. tabaci* Middle East-Asia Minor 1 (MEAM1, former biotype B) cryptic species, which was identified sometime in the late 1980s, principally via the trade in ornamentals, has successfully colonized and spread to at least 54 countries [29,32], and its ability to adapt to various environmental temperatures is one of the key reasons for its success [33,34,35].

Previous studies have shown that exposing whiteflies to thermal stress is accompanied by rapid alteration of gene expression [36,37,38,39,40]. In addition, Lü et al. [41] found that *B. tabaci* significantly improved its survival rate within two generations after heat shock selection experiments and speculated that this rapid increase in viability may be related to epigenetic regulation. Furthermore, studies on DNA methyltransferases involved in epigenetic regulation indicated that DNA methyltransferase 1 and DNA methyltransferase 3 functioned directly in the temperature tolerance of *B. tabaci* [39,42]. These results implied that a possible explanation for rapidly increased temperature tolerance is the regulatory plasticity of genes mediated by epigenetic mechanisms. While ISWI-containing chromatin remodeling complexes were thought to be related to the rapid acquisition of adaptative traits and thermal resistance, relevant experimental evidence is scarce.

In the present study, we assessed the functions of *BtISWI* in invasive MEAM1. First, we cloned the full-length cDNA sequence of the MEAM1 *BtISWI* gene and analyzed its characteristics. Compared with the annotated ISWI from the draft genome of *B. tabaci* MEAM1 [43], the nucleic acid sequence similarity was 78.18%, and the deduced amino acid sequence similarity was 100%. Second, expression profiles of *BtISWI* in response to thermal stress were determined. Third, we identified the function of *BtISWI* in thermal resistance by RNA interference (RNAi). Finally, the temperature preferences of MEAM1 adults after dsRNA feeding were identified. These data will help us to reveal novel functions for the ISWI ATPase in vivo and lay the foundation for further studies aimed at understanding the potential links between chromatin remodeling and thermal plasticity in invasive insects.

## 2. Materials and Methods

### 2.1. Insect Rearing

The *B. tabaci* MEAM1 cryptic species used in this study were reared on healthy cotton plants, *Gossypium hirsutum* (L.) (Zhong No. 287), in insect-proof cages in a greenhouse at 26 ± 1 °C, 50–60% relative humidity (RH), and a photoperiod of 14:10 h (L:D). The cotton plants were individually grown in 9 cm-diameter pots under the same conditions as the whiteflies.

### 2.2. Total RNA Isolation and Reverse Transcription

Total RNA was isolated from the whiteflies using TRIzol (Invitrogen, Carlsbad, CA, US) according to the manufacturer’s instructions. Subsequently, the quality and concentration of RNA were evaluated using a NanoPhotometer^TM^ P330 (Implen, Munich, Germany) and 1% agarose gel electrophoresis. The first-strand complementary DNA (cDNA) was generated from 1.0 µg RNA using the One-Step gDNA Removal and cDNA Synthesis SuperMix (TransGen, Beijing, China).

### 2.3. Cloning and Sequencing of BtISWI

To identify the *BtISWI* gene, BLASTP (protein-protein Basic Local Alignment Search Tool) and TBLASTN (search translated nucleotide databases using a protein query) searches of *B. tabaci* MEAM1 genome information [43] were performed using published ISWI protein sequences from *Drosophila melanogaster* (NP_523719.1), *Nilaparvata lugens* (XP_022189967.1), and *Acyrthosiphon pisum* (XP_001945595.1) as queries. According to the sequence obtained above, specific primers were designed (as shown in Table 1). All PCR (Polymerase Chain Reaction) amplifications were carried out using FastPfu DNA Polymerase (TransGen, Beijing, China), and amplified fragments were purified using an AxyPrep^TM^ DNA Gel Extraction Kit (Axygen, West Orange, NJ, US), then cloned into the pEASY-Blunt Vector (TransGen, Beijing, China) following the manufacturer’s instructions and sequenced.

### 2.4. Bioinformatics Analysis

Sequence alignments and comparisons of sequence similarity were performed using DNAMAN (version 7.0; LynnonBioSoft, Quebec, QC, Canada). ORF FINDER (http://www.ncbi.nlm.nih.gov/orffinder/) was used to predict ORFs. The physical and chemical properties of the deduced amino acid sequence were predicted using ExPASy (http://web.expasy.org/protparam/). Conserved functional domains were identified with the online CONSERVED DOMAINS database of NCBI (https://www.ncbi.nlm.nih.gov/Structure/cdd/wrpsb.cgi). The three-dimensional structures of the protein domains were produced with SWISS-Model (https://www.swissmodel.expasy.org/) and edited using PyMOL. Multiple protein sequence alignments were performed using ClustalW, and the phylogenetic tree was constructed using the maximum likelihood method based on the Whelan Goldman (WAG) model with 1000 bootstrap replications in MEGA7.0 software [44,45].

### 2.5. Real-Time Quantitative PCR

Quantitative real time PCR (qRT-PCR) was used to analyze the gene expression profile with an ABI 7500 real-time PCR system (Applied Biosystems, Foster City, CA, USA). Total RNA extraction and cDNA synthesis were conducted, as described above. The primers used in this study are listed in Table 1, including two reference genes—beta-1-tubulin (β-tub) and elongation factor 1 alpha (EF1-α)—to normalize the mRNA expression levels [46]. Each reaction (20 µL) contained 10.0 µL qPCR SYBR Green Master Mix (Yeasen, Shanghai, China), 1.0 µL cDNA template, 0.4 µL of each primer (10 µM), and 8.2 µL of double distilled H_2_O. The PCR procedure was as follows: 95 °C for 5 min, then 40 cycles of 95 °C for 10 s and 60 °C for 30 s, followed by a melting curve analysis. There were three repetitions for each treatment and control, with 200 adults in each repetition and the ratio of females:males was 1:1. Each repetition was assessed in triplicate (technical replicates).

### 2.6. Temperature Stress Treatment

Based on our previous studies [40,42], we selected 0, 12, 35, and 40 °C for short-term temperature stress treatments. Owing to research found that adult age was associated with different responses to thermal tolerance [47], we standardized adult age using only newly emerged whiteflies that were younger than 3 h. MEAM1 adults were placed together in a 1.5 mL centrifuge tube, then the tubes were placed in a constant-temperature environment (0, 12, 35, and 40 °C) for 1, 3, and 5 h. Adults maintained at 26 °C served as a control group. After thermal stress treatment, whiteflies were collected immediately for qRT-PCR. Each treatment was biologically replicated three times, with 200 adults in each repetition and the ratio of females:males was 1:1.

### 2.7. RNA Interference

Double-stranded RNA (dsRNA) used in this study was synthesized using the MEGAscript T7 High Yield Transcription Kit (Ambion, Austin, TX, USA) with gene-specific primers (Table 1). Then, the *ISWI* dsRNA was diluted to 0.4 µg/µL in a 10% sucrose solution for further feeding treatment (for detailed methods, refer to References [42,48]). Simultaneously, we set control groups including untreated whiteflies, those fed with eGFP-specific dsRNA (dseGFP), and those fed with 10% sucrose. After feeding for 3 h, partial samples were immediately frozen with liquid nitrogen and then stored at −80 °C for analyzing the *BtISWI* mRNA expression level. The rest of the newly emerged MEAM1 adults were collected immediately for thermal tolerance tests or temperature preference behavioral assays.

### 2.8. Thermal Tolerance Test after dsRNA Feeding

For heat resistance, we measured the tolerance times for whiteflies exposed to a high temperature [49,50,51]. Two whiteflies were placed in a 1.5 mL centrifuge tube, then the tube was transferred to a constant temperature of 45 ± 0.2 °C in a water bath, controlled with a CC-106A temperature controller (Huber Kältemaschinenbau GmbH). We measured the interval between the time the tube was placed in the water bath and the time when the whitefly lost control of its body and could not stand autonomously. For cold tolerance, we measured the recovery time following a chill coma induced by cold shock [50,52]. MEAM1 adults, placed in tubes, as described above, were exposed to −5 ± 0.2 °C in a refrigeration bath circulator (K6-cc-NR, Huber Kältemaschinenbau GmbH, Offenburg, Germany) for 10 min, and the recovery time was observed at 26 ± 0.2 °C. Each treatment had four biological replicates, with 200 adults in each repetition and the ratio of females:males was 1:1. The 45 °C and −5 °C conditions were selected based on pre-experiments showing that these temperatures were the discrimination points for whitefly temperature tolerance [50].

### 2.9. Temperature Preference Behavioral Assay

A temperature control device that can generate a series of thermal gradients was used to assay the temperature preference of MEAM1 adults after dsRNA feeding. The temperature gradient (12–30 °C) was produced in an aluminum block (30 cm long × 10 cm wide × 1 cm high) with a slope of 0.72 °C per cm, and thermal probes were embedded in the block every 1.2 cm. There were 20 observation fields for recording the distribution of whiteflies. A 0.1 cm high glass cover confined the whiteflies to the thermal gradients and ensured that adults could migrate freely. Experiments were carried out in dark conditions to prevent the effects of light on temperature preference [53]. Previous studies showed that the state of whiteflies was stable when they moved for 2 h in the darkness. Therefore, in this study, whitefly adults were placed between the aluminum block and the glass cover; they were then allowed to migrate into darkness for 2 h. Finally, the number of whiteflies in each region was counted. The experiments were performed in a room with the temperature maintained at 26 °C. There were four repetitions for each treatment, containing 200 adults in each repetition and the ratio of females:males was 1:1.

### 2.10. Statistical Analysis

The mRNA expression levels were calculated with the 2^−ΔΔCT^ method: [ΔΔCT = (Ct_target_ − Ct_reference_)_treatment_ − (Ct_target_ − Ct_reference_)_control_] [54,55]. Statistical analyses were carried out using the SAS 9.4 software package (SAS Institute, Inc., Cary, North Carolina). We used one-way analysis of variance (ANOVA), followed by Duncan’s multiple range tests to compare the differences between control and treatment groups. Data are presented as mean ± standard error (mean ± SEM). Differences were considered significant when *p* < 0.05.

## 3. Results

### 3.1. Characterization of BtISWI

The complete cDNA of *BtISWI* consists of 3712 nucleotides and contains a 232 bp 5′-UTR (residues 1–232), a 411-bp 3′-UTR (residues 3302–3712), and a 3069 bp ORF (residues 233–3301) (Figure 1). This ORF encodes a predicted protein of 1022 amino acid residues, with a calculated molecular mass of 118.86 kDa and an isoelectric point (pI) of 8.36. Conserved domains analysis of the deduced protein sequence showed that BtISWI contains the N-terminal ATPase domain (DEXDc positions 137–324 and HELICc positions 474–558), and the C-terminal HAND (positions 704–804), SANT (SWI3, ADA2, N-CoR, TFIIIB domains)(positions 805–855), and SLIDE (SANT-like ISWI domain) (positions 890–984) domains (Figure 2A). In addition, we used homology modeling in the Swiss-model to further refine and confirm the above predictions. BtISWI had similar functional domains compared to *Drosophila melanogaster* (the similarity was 85.39%, QMEAN (Qualitative Model Energy Analysis) was 0.14 and GMQE (Global Model Quality Estimation) was 0.74) (Figure 2B); the HAND domain has four helices (H1, H2, H3, and H4), the SANT domain contains three helices (SA1, SA2, and SA3), and the SLIDE domain also consists of three helices (SL1, SL2, and SL3). The SANT and SLIDE domains are connected by a continuous α-helix (SP). Moreover, multiple sequence alignment (Figure 3) and phylogenetic analyses (Figure 4) indicated that insect *ISWI* genes were highly conserved throughout evolution, and the ISWI proteins of insects in the same order clustered on the same branch of the phylogenetic tree.

### 3.2. Expression Profiles of BtISWI in Response to Thermal Stress

qRT-PCR was carried out to investigate whether thermal stress affects the expression profiles of *BtISWI*. The results showed that, compared to the whiteflies maintained at 26 °C (the control temperature), mRNA expression levels of MEAM1 *BtISWI* were significantly up-regulated after exposure to heat shock or cold shock conditions (except for those exposed to 12 °C for 1 h) (Figure 5). These data suggested that the expression of *BtISWI* was affected by thermal stress.

### 3.3. BtISWI mRNA Expression Level after Feeding dsRNA

In order to further explore the function of *BtISWI*, we fed MEAM1 adults with dsRNA to silence *BtISWI* gene expression. Compared with controls—whiteflies that were untreated or those fed with dsRNA targeting enhanced GFP (ds*eGFP*) or 10% sucrose—the *BtISWI* mRNA expression level was decreased by 43.10% after feeding with *dsISWI* for 3 h (Figure 6).

### 3.4. The Role of BtISWI in Temperature Stress

The thermal tolerance of MEAM1 adults fed with dsRNA targeting *BtISWI* was significantly reduced, manifested by a shorter heat shock tolerance time (Figure 7A) and a longer chill coma recovery time (Figure 7B). The mean tolerancetimes after 45 ± 0.2 °C heat shock for whiteflies that were untreated or fed with 10% sugar, ds*eGFP*, or ds*BtISWI* were 26.51 ± 1.06 min, 26.23 ± 0.38 min, 26.86 ± 0.84 min, and 16.12 ± 1.32 min, respectively. The mean chill coma recovery times (Figure 7B) after 10 min exposure to −5 ± 0.2 °C for MEAM1 individuals that were untreated, or those fed with 10% sugar, ds*eGFP*, or ds*BtISWI* were 6.11 ± 0.32 min, 6.28 ± 0.19 min, 6.25 ± 0.37 min, and 8.99 ± 0.13 min, respectively. These results suggested that *BtISWI* might be essential in the thermal resistance of MEAM1 adults.

### 3.5. The Choice of Preferred Temperature

In order to examine whether the temperature preference of MEAM1 adults changed after dsRNA feeding, a linear thermal gradient ranging from 12 to 30 °C was tested. The results showed that the preferred temperature for untreated whiteflies was 13.8 °C (Figure 8), with the percentage of whiteflies (21.98 ± 4.49%) in this area significantly exceeding those of the other temperature regions. The preferred temperature for whiteflies fed 10% sugar or fed ds*eGFP* was consistent with the untreated whiteflies (the percentages of preferred temperature region were 19.60 ± 2.02% and 19.76 ± 3.13%, respectively). When the temperature preference was tested in individuals subjected to ds*ISWI* feeding, they exhibited temperature preferences of 15.7 °C and 17.3 °C, with percentages of whiteflies in these two areas of 21.50 ± 3.49% and 20.36 ± 3.55%, respectively. ANOVA analyses showed that the distribution of whiteflies in these two regions significantly exceeded others, but there was no significant difference between them (Figure 8). These results implied that the temperature preference of MEAM1 changed after ds*ISWI* feeding.

## 4. Discussion

In this study, the full cDNA sequence of the *ISWI* gene in invasive MEAM1 was reported. We found that BtISWI had the same structural organization with ISWI identified previously in other species: A highly conserved ATPase domain in the N-terminal half of the ISWI protein and a representative HAND–SANT–SLIDE domain in the C-terminal half of the protein (Figure 2A) [56]. Research on ISWI structural domains has shown that efficient remodeling depends on the presence of the C-terminal HAND–SANT–SLIDE domain [57,58]. Remarkably, the C-terminal end of MEAM1 shared 74.32% sequence similarity with other insects, indicating that the function of ISWI is relatively conserved across species. In addition, the phylogenetic tree showed that ISWI proteins of insects in the same order clustered on the same branch (Figure 4), consistent with traditional taxonomy.

The small invasive insect MEAM1, which is thought to originate in the Middle East–Asia Minor region, has successfully colonized and spread to at least 54 countries [29,59], mainly due to its rapid adaption to various temperature conditions [35,60,61]. In the present study, the mRNA expression level of MEAM1 *BtISWI* was significantly upregulated after adults were exposed to high or low temperatures, up to the highest level after exposure to 35 °C for 5 h (Figure 5). Interestingly, for the same high temperature stress conditions, previous research found that the onset temperatures for the synthesis of *hsp20*, *hsp70*, and *hsp90* were 35 °C, 39 °C, and 35 °C, respectively [62]. These data showed that the onset temperature for *hsp20* and *hsp90* expression was consistent with the temperature for inducing *BtISWI* expression up to the highest level, and the onset temperature for *hsp70* expression was generally 4 °C higher. Furthermore, studies in *Drosophila* showed that NURF, a chromatin-remodeling complex that contains ISWI as its ATPase subunit, cooperated with the GAGA factor to mobilize nucleosomes on the promoter of the heat-shock genes, creating a nucleosome-free domain over the promoter, thus exposing suitable sites for the heat-shock TF [21,63]. Collectively, these results indicated that *BtISWI* might improve the ability of MEAM1 individuals to adapt rapidly to various temperature conditions by inducing the expression of stress-related genes, such as heat-shock protein genes.

ISWI-containing chromatin remodeling complexes function directly in the thermal adaption of organisms. For example, Erkina et al. [23] demonstrated that combinatorial inactivation of ISWI and SNF2 had a strong synergistic effect by diminishing histone loss during heat induction and eliminating Pol II recruitment. Importantly, this inactivation also eliminated preloading of HSF promoters before heat shock, which is the primary regulator of most HSP genes [23,64]. In addition, the generation of an accessible heat shock promoter in chromatin in *Drosophila* required the concerted action of the GAGA TF and the ISWI-containing complex (dNURF) [21,63]. In this study, the functions of the MEAM1 *BtISWI* in thermal tolerance were identified using RNAi. Results showed that the mean heat tolerance time of MEAM1 adults fed *BtISWI* dsRNA was significantly decreased and the mean chill coma recovery time was significantly increased (Figure 7). The mechanisms of the response to high-temperature tolerance and low-temperature chill coma recovery are, in some respects, probably closely related to the thermal stress response [65]. Hence, it is reasonable to conclude that *BtISWI* has important effects on the thermal tolerance of MEAM1. However, only the common subunit of ISWI was researched in the present study, so further study is needed to determine the specific ISWI complexes essential for the thermal tolerance of whiteflies.

Due to a temperature preference rhythm, (for example, the preferred temperature in *Drosophila* rises during the day and falls during the night) [66,67], our experiments were conducted during the same periods of one day. Compared with *D. melanogaster* or *Apolygus lucorum*, whose preferred temperature was consistent with their optimum temperature range [68,69], whitefly adults exhibited a lower temperature preference than their optimum temperature. Similarly, Kinzner et al. [70] also found that adults of the tropical *D. birchii* had a low temperature preference of 16.7 °C. The preferred temperature is theoretically expected to be in a similar range than the optimum temperature of maximum physiological performance, but empirically, the preferred temperature can be lower than the optimum temperature [70,71]. Furthermore, in the present study, after feeding *BtISWI* dsRNA, the preferred temperature of the MEAM1 adults was about 1.9–3.5 °C higher than the control groups (Figure 8). In addition, whiteflies fed *BtISWI* dsRNA showed a wider range of favorable temperatures than controls. Organisms that are good at thermoregulation may distribute across a relatively narrow range of favorable temperatures [72,73,74]. Therefore, the results of this study indicated that the *ISWI* gene might be involved in the temperature regulation of the invasive insect *B. tabaci* and affect its temperature preference behavior.

## 5. Conclusions

Previous studies on ISWI have focused on model organisms, such as *Drosophila*, yeast, nematodes, mouse, and *Arabidopsis thaliana*. This study was the first to reveal the characteristics of *BtISWI* in invasive *B. tabaci* MEAM1 and to identify the functions of *BtISWI* using RNAi. Moreover, the temperature preference behavior of whiteflies was first measured in the present study. The results of this study demonstrated that *BtISWI* played an essential role in thermal tolerance and may participate in the temperature regulation of MEAM1. Our findings highlighted the importance of epigenetic gene regulation in the thermal response and thermal adaptation of invasive insects. Furthermore, this study provides a new potential target to establish sustainable control strategies for insect pests.

## Figures and Tables

**Figure 1 insects-11-00113-f001:**
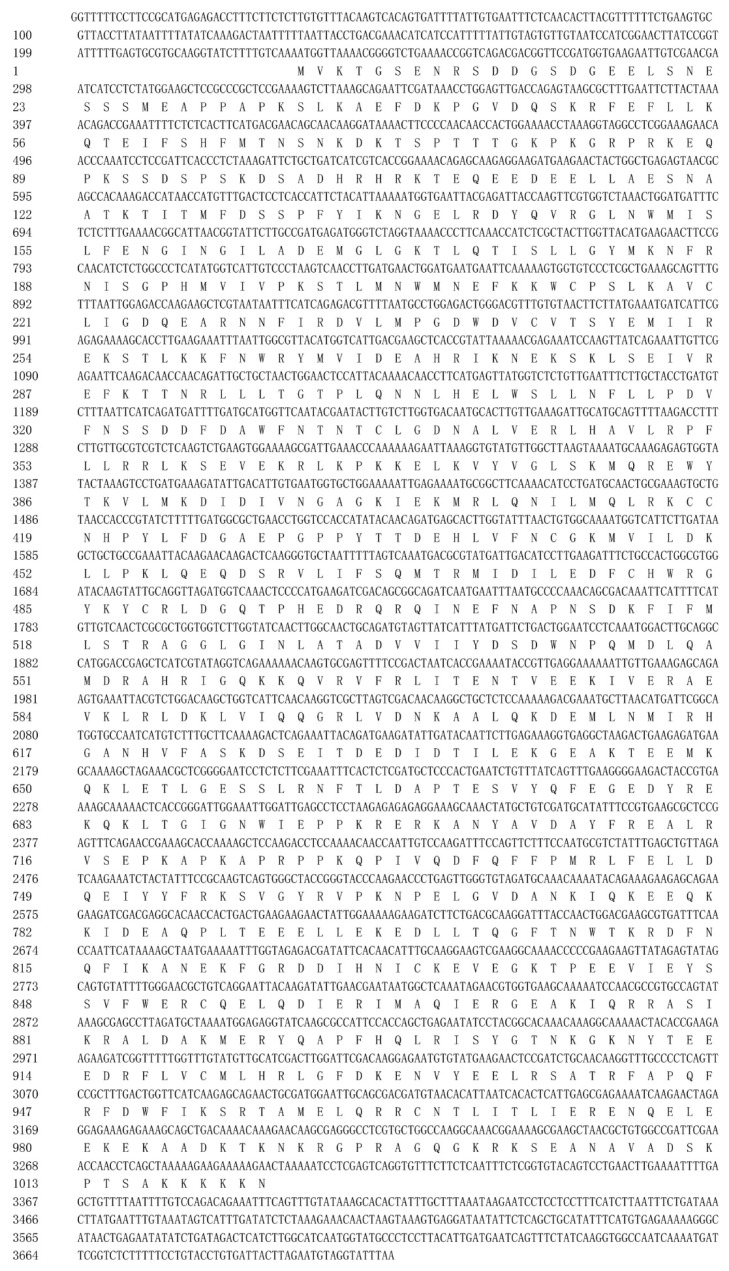
This The full-length cDNA sequence of *Bemisia tabaci* MEAM1 *BtISWI* and its deduced amino acid sequence. The full-length cDNA of MEAM1 *BtISWI* is 3712 bp, and the ORF (233–3301 bp) encodes a polypeptide of 1022 amino acids.

**Figure 2 insects-11-00113-f002:**
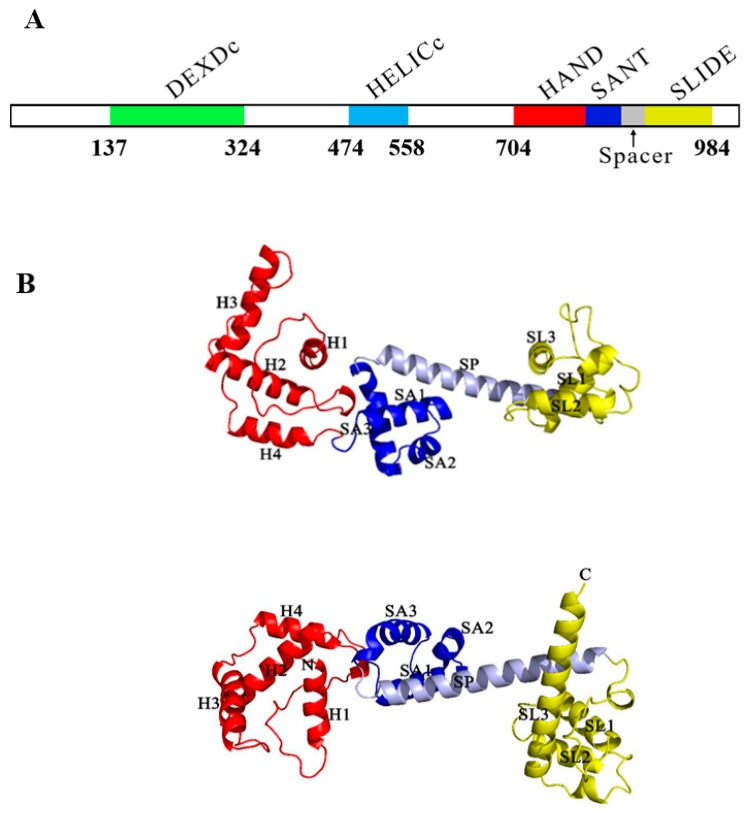
Functional domains in *Bemisia tabaci* MEAM1 BtISWI protein (**A**) and two orthogonal views of the predicted BtISWI C-terminal structure (**B**). The DEXDc domain is depicted in green, HELICc domain in baby blue, HAND domain in red, SANT (SWI3, ADA2, N-CoR, TFIIIB) domain in blue, SLIDE (SANT-like ISWI) domain in yellow, and the spacer helix connecting the SANT and SLIDE domains in gray.

**Figure 3 insects-11-00113-f003:**
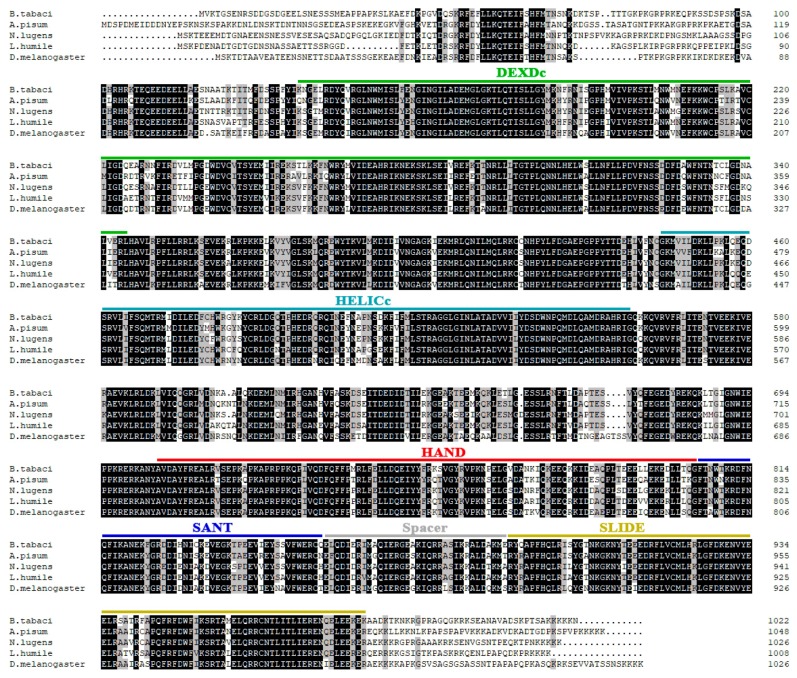
Multiple alignments of the imitation switch (ISWI) protein sequences from *Bemisia tabaci* MEAM1 and other insects. The conserved domains are highlighted in the same colors as in Figure 2. Multiple sequence alignment revealed that the deduced amino acid sequence of BtISWI is highly conserved when compared to previously identified ISWI amino acid sequences. *B. tabaci*: *Bemisia tabaci*; *A. pisum*: *Acyrthosiphon pisum* (XP_001945595.1); *N. lugens*: *Nilaparvata lugens* (XP_022189967.1); *L. humile*: *Linepithema humile* (XP_012220642.1); and *D. melanogaster*: *Drosophila melanogaster* (NP_523719.1).

**Figure 4 insects-11-00113-f004:**
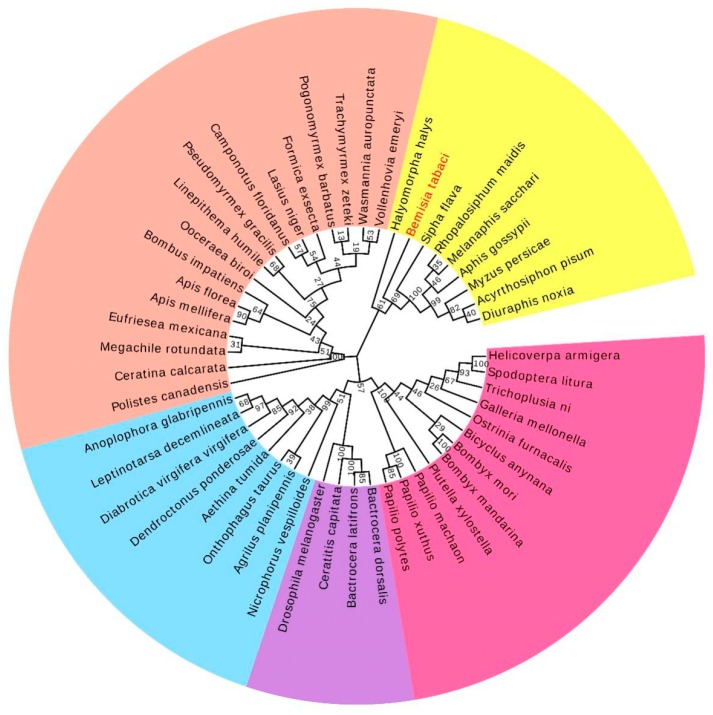
A phylogenetic tree based on the known amino acid sequences of ISWI genes. The phylogenetic tree was constructed using the maximum likelihood method with 1000 bootstrap replications in MEGA 7.0 software. Numbers at the nodes of the branches represent the level of bootstrap support for each branch. Insects in the same order clustered on the same branch. Lepidoptera is depicted in pink, Diptera in purple, Coleoptera in baby blue, Hymenoptera in dusty pink, and Hemiptera in yellow. Appendix A shows the ISWI protein sequences used for phylogenetic analysis.

**Figure 5 insects-11-00113-f005:**
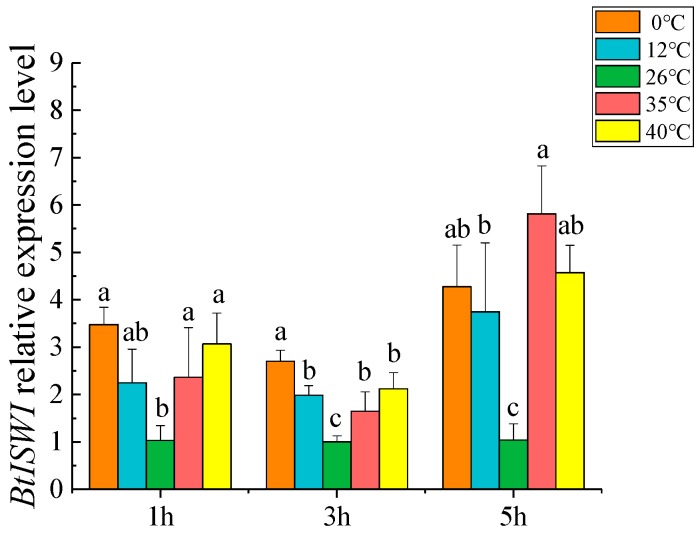
Relative expression levels of MEAM1 *BtISWI* after exposure to different temperatures. *BtISWI* mRNA expression was significantly up-regulated (except for exposure to 12 °C for 1 h) after exposure to heat shock and cold shock compared to the controls (26 °C, and the *BtISWI* mRNA expression at this temperature was stable all the time). The relative expression levels are expressed as the mean ±SEM. Bars with different lowercase letters are significantly different at *p* < 0.05.

**Figure 6 insects-11-00113-f006:**
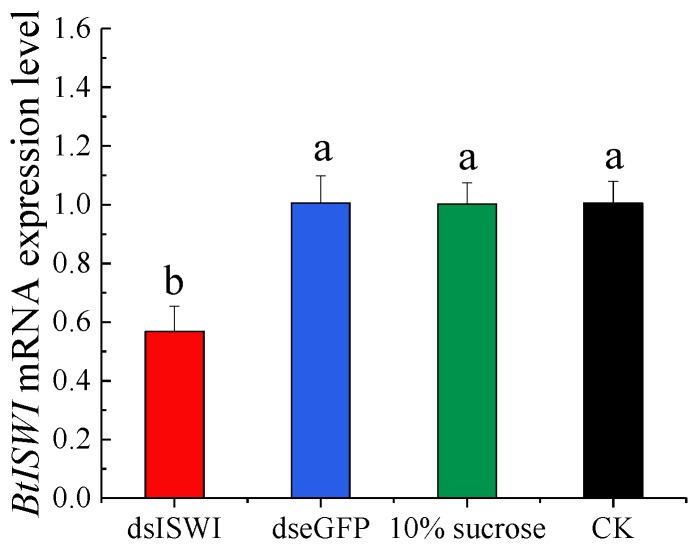
Effect of double-stranded RNA (dsRNA) treatment on *Bemisia tabaci* MEAM1 *BtISWI* mRNA expression level. *BtISWI* mRNA expression in whiteflies fed *BtISWI* dsRNA (dsISWI) for 3 h was significantly decreased compared to the controls: whiteflies untreated or fed eGFP dsRNA (dseGFP) or 10% sucrose. The expression levels are expressed as the mean ±SEM. Bars with different lowercase letters are significantly different at *p* < 0.05.

**Figure 7 insects-11-00113-f007:**
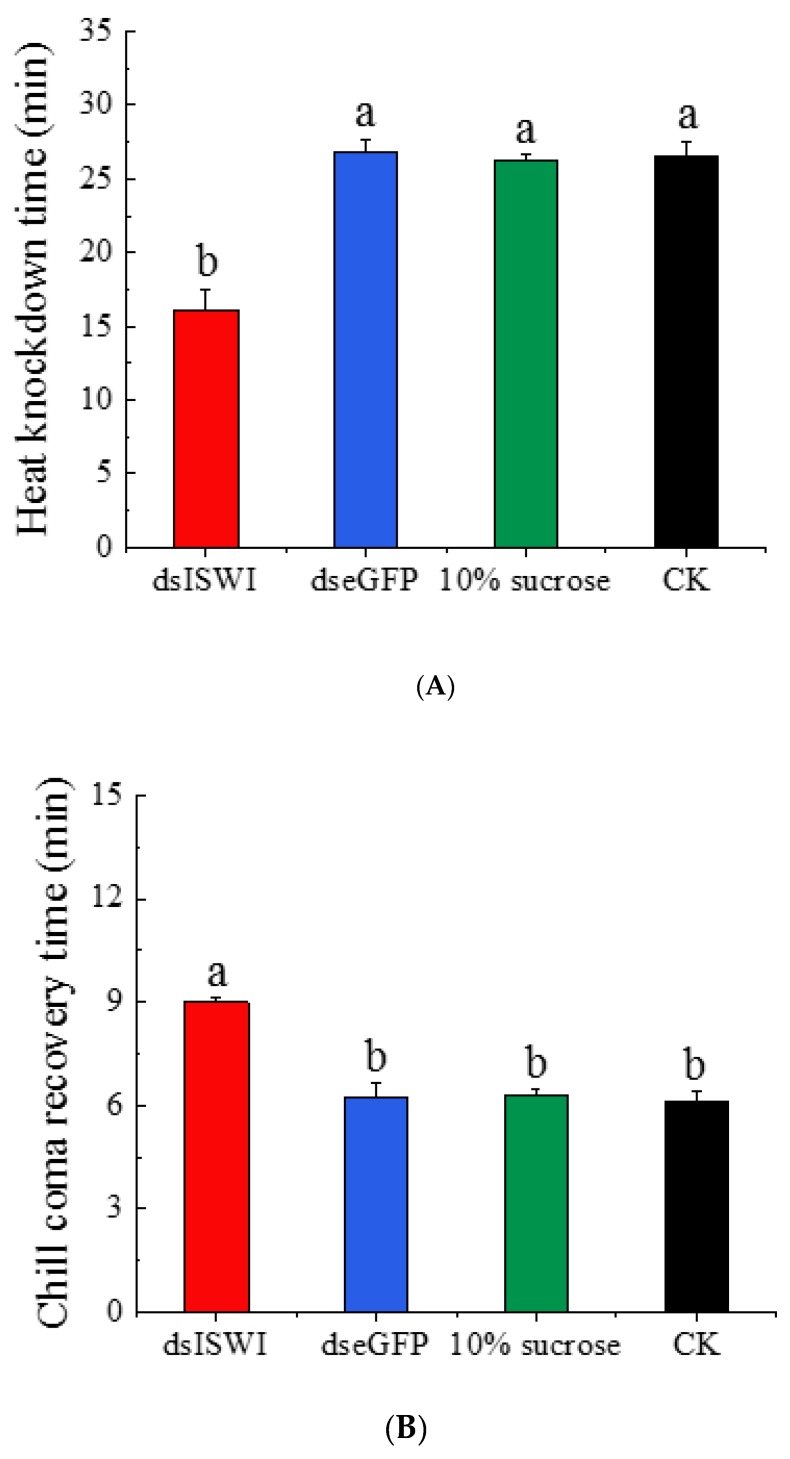
Mean heat tolerance time and chill coma recovery time after double-stranded RNA (dsRNA) feeding in MEAM1 adults. Compared with the control groups (whiteflies untreated or fed dseGFP or 10% sucrose), the mean heat tolerance time for whiteflies fed dsISWI was significantly decreased (**A**), whereas the mean chill coma recovery time was significantly increased (**B**). The data are presented as the mean ±SEM; N = 200. Bars with different lowercase letters are significantly different at *p* < 0.05.

**Figure 8 insects-11-00113-f008:**
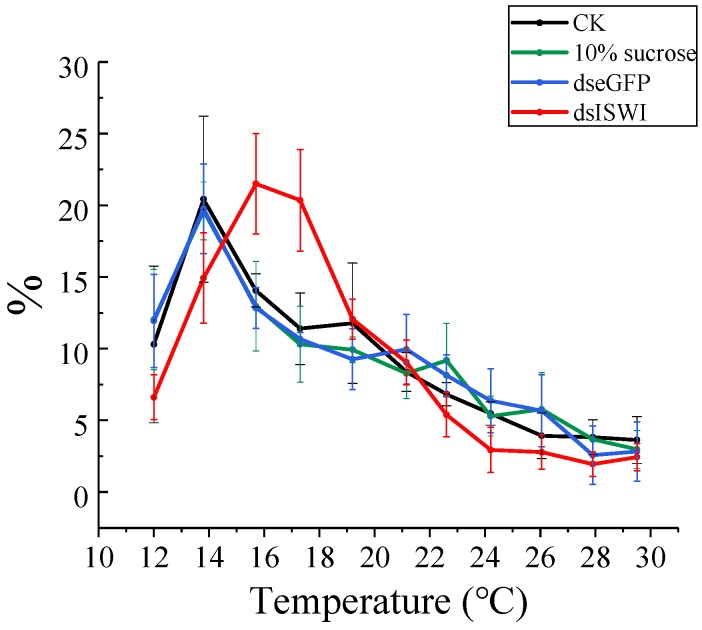
Temperature preference response of MEAM1 adults. The number of whiteflies in each biological replicate was 200, and there were four repetitions for each treatment. Compared to the control treatments (whiteflies untreated, or fed dseGFP or 10% sucrose), whiteflies fed dsISWI showed a higher preferred temperature and a wider distribution range of favorable temperatures. The data are presented as the mean ±SEM.

**Table 1 insects-11-00113-t001:** Primers used for cDNA cloning, qRT-PCR (Quantitative real time PCR), and dsRNA synthesis.

Primer	Primer Sequence (5′ to 3′)
Primers for full-length gene amplification	
ISWI-F1	GCTTCAGCCAATATGGCGACT
ISWI-R1	CGGCAGCAGCTTATCAAGAATG
ISWI-F2	GACCTTTCTTGTTGCGTCGTCTC
ISWI-R2	GGCGCTTGATACCTCTCCATTT
ISWI-F3	ACTATTTCCGCAAGTCAGTGGGC
ISWI-R3	GACGACGCATCTCAAGGCTAAA
Primers for real-time quantitative PCR	
ISWI-QF	GCAGGTTAGATGGTCAAACTCCCC
ISWI-QR	TTTTCCTCAACAGTATTTTCGGTG
EF1-α-F	TAGCCTTGTGCCAATTTCCG
EF1-α-R	CCTTCAGCATTACCGTCC
β-tub-F	TGTCAGGAGTAACGACGTGTTTG
β-tub-R	TTCGGGAACGGTAAGTGCTC
Primers for dsRNA synthesis	
ISWI-DF	TAATACGACTCACTATAGGGCTCCGATTCACCCTCT
ISWI-DR	TAATACGACTCACTATAGGGGTCCCAGTCTCCAGGC

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
