# Peer review of "Using RNA Interference to Reveal the Function of Chromatin Remodeling Factor ISWI in Temperature Tolerance in Bemisia tabaci Middle East–Asia Minor 1 Cryptic Species"

_insects, 2020, doi:10.3390/insects11020113_

Round 1

Reviewer 1 Report

The authors present a well written, scientifically sound manuscript that details their studies regarding the chromatin remodelling protein ISWI of B. tabaci.  The authors cloned the putative gene, conducted sequencing, alignment with homologues of other insects, conducted in silico modelling of the ISWI and constructed a phylogenetic tree with respect to ISWI genes of insects from several orders. All sound preliminary investigation

Expression analysis of the B. tabaci ISWI gene was subsequently conducted with respect to temperature. The authors showed up regulation with respect to temperature either below or above the ideal temperature of 26degC  establishing that the ISWI gene may function in hot and cold adaption.

A putative RNAi effect was observed when B. tabaci was feed dsRNA targeting the ISWI gene. Although significant, I would expect that the knockdown be more prominent and the statement of the authors ‘knocked down by half’ (Line 244) is misleading in that there is no figure (#) or percent (%) and the graph clearly shows it is closer to 0.6 than 0.5 expression level. This should be amended.

The authors then functionally investigate the effects of knockdown of the ISWI gene on thermal and cold tolerance. Here the science is sound however, the explanation the heat tolerance is confusing. The nomenclature used here (ie. ‘heat shock knockdown time’) has precedent from ref 49 but is confusing if one is not familiar with the context and ‘knockdown’ is generally used to refer gene ‘knockdown’ following and RNAi event. May I suggest the use of ‘tolerance’ in place of ‘knockdown’ (eg LINE 253 + :eg. … manifested by a shorter heat shock tolerance time’) using this word will aid in readability and contrast the opposite and simply explained and understood ‘cold tolerance’.

Line 153: replace ‘knockdown’ with ‘tolerance’

Line 244: include a figure (#) or a percent figure to establish the amount of knockdown.

Line 253: replace ‘knockdown’ with ‘tolerance’

Figure 7: graph A : replace ‘knockdown’ with ‘tolerance’

Reviewer 2 Report

In the present study, the author characterized the imitation SWI (ISWI) from Bemisia tabaci MEAM1 and further validate its role in the thermal response by RNAi. The study is well written but the grammar is sometimes not correct and articles are missing quite often, so I recommend the article should be checked by a native speaker. The experiments clearly demonstrate the role of ISWI in temperature tolerance in Bemisia tabaci. However, controls are not described in detail and statistical analysis needs to be redone (see comment below). Therefore, I recommend accepting the manuscripts after minor changes:

Introduction:  

The author should add about the draft genome of whitefly Bemisia tabaci MEAM1 (reference no. 43) and also compare their cloned BtISWI with other previously annotated ISWI from the draft genome.

Results:

It is unclear from the method section (line 146) which concentration of dsRNA is used for experiments. So, the author should clearly mention the concentration of dsRNA used in each experiment (section 3.3, 3.4 and 3.5) Inline 196-197, Author should mention the structural similarity (%) and QMEAN value of BtISWI Model In Figure 2A, either the domain position should be labeled or I recommend to redraw the gene structure by using MyDomains - Image Creator available on expasy. (https://prosite.expasy.org/mydomains/) In Figure 4; Line 232, pink color was mentioned to depict both Lepidoptera and Hymenoptera, the author should correct the depicted color of each insect order in the figure legend.    In section 3.2, Author shows that the expression levels of BtISWI were significantly upregulated after thermal changes, however they did not mention the expression levels of BtISWI at control temperature (26 °C) during different time interval (1, 3, 5 hrs), whether it significatly changed or not? It becomes further confusing to understand in figure 5, (line 238) where author mentioned “different lowercase letters are significantly different at P < 0.05”   In Figure 8, Author should redo the statistical analysis, it becomes difficult to understand because there were no lowercase letters in the graph and it does not even have any bar in the graph.  
